# Determinants of Vitamin D Supplementation among Individuals with Type 1 Diabetes

**DOI:** 10.3390/ijerph17030715

**Published:** 2020-01-22

**Authors:** Mikołaj Kamiński, Magdalena Molenda, Agnieszka Banaś, Aleksandra Uruska, Dorota Zozulińska-Ziółkiewicz

**Affiliations:** Department of Internal Medicine and Diabetology, Poznan University of Medical Sciences, ul. Mickiewicza 2, 60-834 Poznań, Poland; magdalena.molenda98@gmail.com (M.M.); agula.banas@gmail.com (A.B.); aleksandrauruska@gmail.com (A.U.); zozula@box43.pl (D.Z.-Z.)

**Keywords:** vitamin D, survey, supplementation, type 1 diabetes, diabetology, diabetologist, Poland, dietary supplement, social environment, physician

## Abstract

Half of the individuals with type 1 diabetes (T1DM) may present Vitamin D (VD) deficiency. There is little known about factors determining a decision on VD supplementation. The study aimed to determine the factors affecting vitamin D supplementation in people with T1DM. A cross-sectional survey study using the authors’ questionnaire paper and its digital version was performed. The questions involved data on the basic characteristics of the respondent, medical history, VD supplementation status, influence of the social environment, self-education, and the most important personal motivator for VD supplement use. Multivariate logistic regression analysis was performed. We collected a total of *n* = 184 papers and *n* = 550 digital complete surveys. From 734 total respondents, 62.0% declared VD supplementation. The main personal rationale for VD supplementation were recommendation of medical specialist 172 (37.8%) and self-education 135 (29.7%). The main reasons for non-supplementation of VD were lack of knowledge about VD 159 (57.0%) and lack of motivation 77 (27.6%). VD supplementation was independently associated with a family doctor (odds ratio (OR), 95% confidence interval (CI): 4.67, 2.32–9.40) or medical specialist recommendation (16.20, 9.57–27.43), and self-education (5.97, 3.90–9.13). Most Polish individuals with T1DM use VD supplements, and the decision is related to physicians’ recommendations and self-education.

## 1. Introduction

Vitamin D (VD) is a prohormone produced by the skin in response to sunlight stimulation and ingested with food (e.g., milk, dairy products, and fish) [1]. The active form of VD is calcitriol (1,25-dihydroxycholecalciferol (1,25(OH)2D)), however, this form is characterized by relatively short biological half-time (4–6 h) and low concentration in serum (measured in pg/mL or pmol/L), while prohormone, cholecalciferol (25-hydroxycholecalciferol (25(OH)D)) has a higher serum level (measured in ng/mL or nmol/l) and longer biological half-time (2–3 weeks) than calcitriol [2,3]. For this reason, cholecalciferol is used for both assessments of VD status and supplementation of VD [3,4,5,6]. The primary role of VD is a regulation of calcium and phosphorus homeostasis [7]. Moreover, VD was reported to have pleiotropic effects such as regulation of immunological system [8,9], effects on the cardiovascular system via the renin–angiotensin–aldosterone system [10,11], and regulation of insulin secretion [12,13,14]. In some countries, food is fortified for VD [15]. For instance, in Poland margarine is fortified for VD. Nevertheless, the VD deficiency occurs in 66% of Polish general population [4]. Currently, adults and adolescents are recommended to supplement VD from October through April, while for seniors (>65 years) it is throughout the year [16]. Type 1 diabetes (T1DM) is an autoimmune disease. The autoinflammatory process leads to the destruction of beta-cells of the pancreas, and in consequence to absolute endogenous insulin deficiency. VD deficiency and VD receptor polymorphisms were previously reported to be potential risk factors of developing T1DM [17,18]. Moreover, our observation revealed that VD serum concentration is negatively associated with insulin resistance in adults with T1DM [19]. In the same study, Kamiński et al. reported that 47% of adults with T1DM had VD deficiency (25(OH)D) < 20 ng/mL) and 15% had VD severe deficiency (25(OH)D) < 10 ng/mL)in the period of limited insolation in Poland. Wierzbicka et al. observed that 82% of Polish adolescents with T1DM had VD deficiency, and 25% had severe deficiency [20]. Similarly, in previous studies, more than 75% of patients with T1DM had VD deficiency [21,22]. These reports urge to widely counsel VD supplement use to increase VD serum concentration in populations with T1DM. However, there is little known on factors leading to the decision to use VD supplements or not. The analysis of such conditions may reveal a group of patients who are at risk of insufficient VD supplementation and in consequence VD deficiency.

The aim of the study was to determine the factors affecting vitamin D supplementation in people with T1DM.

## 2. Methods

A survey in the T1DM population using a simple, anonymous, authors’ questionnaire was performed. The questionnaire contained mostly closed-ended questions that were inspired by previous survey studies on VD supplementation [23,24,25]. The questions were discussed by two authors (M.K. and A.U.) to include only essential data points in the final questionnaire (Appendix A). The questions involved data on basic characteristics of the respondent, diabetic history, presence of chronic complications, comorbidities, declaration of VD supplementation (“Do you take vitamin D supplements?”), influence of the social environment (family doctor, medical specialist, relative, or friend recommendation; an admission that relative or friend supplements or no VD), self-education on VD from Internet/media/books (“I learned about the need of vitamin D supplementation from the Internet/media/books”), and the most important personal motivation on VD supplementation (e.g., “Please choose what motivated you the most to begin vitamin D supplementation”) (Appendix A). The term “medical specialist” refers to a physician with 4.5–6.5 years of postgraduate specialization training. The survey was performed in Poland, in the local language. The survey is short, contains mostly categorical, closed-ended questions and does not calculate standardized outcome measurement. Therefore, the questionnaire did not require statistical validation. The responders were requested for feedback and were encouraged to ask any questions regarding the survey. The questions of the responders were related to the deadline to fulfill the questionnaire, when and where results will be published, etc. None of the respondents declared difficulties with understanding the questions from the survey. The Polish and English versions of the questionnaire are presented in Appendix A. The inclusion criteria involved: individuals with type 1 diabetes aged at least 16 years. Incomplete surveys and responses from people with other types of diabetes were excluded from the analysis.

For the study, patients in the Department of Diabetology and the outpatient clinic were requested to fill in a paper version of the questionnaire. The data from paper surveys were prescribed to the REDCap database [26]. Furthermore, we prepared a digital version of the survey in REDCap and the links to the survey were spread on Facebook groups dedicated to Polish diabetics. Information about each group is presented in Appendix A. Data were collected from October 2018 through April 2019. Body mass index (BMI) was counted based on the declared weight and height. Overweight was defined as BMI ≥ 25 kg/m^2^ and below 30 kg/m^2^, and obesity as BMI ≥ 30 kg/m^2^.

Additionally, a short, anonymous authors’ questionnaire dedicated to diabetologists or residents of diabetology was prepared. The questions were inspired by previous survey studies [27,28]. The survey contained questions on personal VD supplementation status, motivation, and whether the respondent recommends patients VD supplementation (Appendix A). A link to the digital survey was sent via e-mail to diabetologists associated with the local branch of Diabetes Poland.

Statistical analysis was performed with STATISTICA 12.0 (StatSoft, Round Rock, TX, USA). The number of positive responses (“YES”) for each respondent was calculated, (The normality of the distributions of the variables was tested using Kolmogorov–Smirnov’s test with Lilliefors correction. Due to a lack of normality, non-parametrical tests were performed. The data are presented as medians (interquartile ranges) or numbers (percentages). To compare the groups who declared or denied VD supplementation and the individuals who filled in the paper vs. the digital version of the survey, the Mann–Whitney U test for numerical data and Pearson chi-square test for categorical variables were used. Each category of the variables with more than two categories was compared separately using the chi-square test. For instance, living in a village vs. living outside of a village. The significance level was set at a *p*-value < 0.05. 

To assess factors that could be associated with VD supplement use univariate and multivariate logistic regression analyses were performed. The dependent variable was declared VD supplementation (count as one) or denied VD supplementation (coded as null). In univariate logistic regression model, the independent factors were: version of the survey (digital coded as one), sex (female coded as one, men coded as null), age, diabetes duration, overweight or obese (coded as one, non-obese coded as null), living place (village coded as one, city counts as null), at least one diabetic complication (coded as one), presence of diabetic complications, (retinopathy, nephropathy, neuropathy, diabetic foot syndrome, ischemic heart disease), concomitant disease (hypothyroidism, coeliac disease, asthma), recommendation and influence of the social environment (answer “Yes” coded as one). The version of the survey was analyzed using a logistic regression model to take into account potential differences between both groups (respondents from the department vs. Facebook users) in clinical characteristics and VD supplementation. Therefore, the version of the survey reflects the data source. For the multivariate logistic regression model, variables with a *p*-value < 0.1 in the univariate regression analysis were chosen. Plots were generated using the “forestplot” package of R 3.5.1 (Vienna, Austria) [29].

## 3. Results

### 3.1. General Characteristics

In total, 1181 (350 paper version, 831 digital) responses were collected, but 734 surveys were completed including 184/734 (25.1%) paper and 550/734 (74.9%) digital surveys. The median age of the study population was 31 (24–39) years (the youngest respondent was 16 years old, and the oldest was 68). The VD supplementation was declared by 455 (62.0%) of all individuals (Table 1).

### 3.2. Comparison of the Study Groups

In the comparison of the group who declared the VD supplementation with the participants who denied VD supplement use, a significant difference in the ratio of paper survey respondents, prevalence of individuals living in village; living in a city of above 50,000 citizens; presence of retinopathy; presence of hypothyroidism; supplementation recommendation made by family doctor, medical specialist, pharmacist, relative; knowledge about VD supplementation acquired from Internet/media/books; ratio of respondents who admitted that their relative supplements VD; friend supplements VD; and number of positive responses on social environment influence were found (Table 1). 

Additionally, the groups who filled in the paper or the digital version of the survey were compared (Appendix A). Both groups differed in ratio of respondents who declared VD supplementation; ratio of females, age; ratio of respondents with age > 65 years; diabetes duration; ratio of respondents living in village, in city > 50,000 of citizens; presence of at least one diabetic complication; ratio of respondents with declared retinopathy, nephropathy, diabetic foot syndrome, ischemic heart disease, hypothyroidism; ratio of positive answers on social environment influence: VD supplementation recommendation made by family doctor, specialized doctor, and number of positive responses on the questions.

### 3.3. Personal Reasons to Use or Not Use Vitamin D Supplementation

In the group who declared VD supplementation, the subjective reasons for VD supplement use were: recommendation of medical specialist 172 (37.8%), knowledge on VD acquired from Internet/media/books 135 (29.7%), relative recommendation 43 (9.5%), family doctor recommendation 38 (8.3%), friend recommendation 27/455 (5.9%), pharmacist recommendation 6 (1.3%), and others 34 (7.5%). Among the other rationales, the most frequent were results of lab findings 20 (4.4%), disease 11 (2.4%). In the group who denied VD supplementation, the most important personal reason for non-supplementation of VD were: lack of knowledge about VD 159 (57.0%), lack of motivation 77 (27.6%), believe that VD has low importance for health 14 (5.0%), consider VD supplementation as a waste of money 11 (3.9%) and others 18 (6.5%). 

### 3.4. Period and Dose of VD Supplementation

In the group who declared VD supplementation, 129 (28.4%) supplemented VD from October to April, 108 (23.7%) irregularly, and 218 (47.9%) throughout the year. There were identified a total of 8 (1.8%) individuals who supplemented VD less than 800 UI/daily (UI/d), 72 (15.8%) from 800 to 1000 UI/d, 180 (39.6%) from 1200 to 2000 UI/d, 96 (21.1%) from 2200 to 4000 UI/d, 23 (5.1%) from 4500 to 8000 UI/d, 5 (1.1%) from 10,000 to 12,000 UI/d, 7 (1.5%) from 13,000 to 80,000 UI/d. Substantial number of the responders (*n* = 64 (14.1%)) did not disclose the VD supplementation dose. Two hundred and fifty-eight (35.1% of total number, 56.7% of subjects who declared VD supplementation) respondents supplement VD in accordance with the current guidelines: adequate for age and BMI. Proper VD supplement use was identified in 27 obese individuals (32.5% of total number of obese subjects, 47.3% of obese subjects who declared VD supplementation) and in 2 respondents aged >65 years (18.2% of total number, 40.0% of subject who declared VD supplementation). 

### 3.5. Logistic Regression Analysis

The results of univariate regression analysis are presented in Appendix A. In univariate regression analysis with dependent variable: declaration of VD supplementation, the condition of *p*-value < 0.1 in group of respondents of paper version of the survey met following variables: version of the survey, sex, living in village, hypothyroidism, family doctor, medical specialist, pharmacist, and relative recommendations, self-education, VD supplementation by relative or friend (Appendix A). In the multivariate regression analysis, VD supplement use was independently associated with the version of the survey (odds ratio (OR), 95% confidence interval (CI): 2.14, 1.37–3.33), family doctor (OR, 95% CI: 4.67, 2.32–9.40) or medical specialist recommendation (OR, 95% CI: 16.20, 9.57–27.43), and self-education (OR, 95% CI: 5.97, 3.90–9.13) (Figure 1). 

### 3.6. Diabetologists Survey

The digital survey was sent to *n* = 93 diabetologists. Answers from 38 (41%) physicians mostly female were obtained. VD supplementation was declared by *n* = 30 (79%) of the respondents. The main motivation for VD supplementation was the knowledge obtained from media/books/Internet (*n* = 26), the counsel of specialist (*n* = 3), and own VD serum concentration (*n* = 1). Most of the respondents declared VD supplementation from October to April (*n* = 19) next to all year (*n* = 7) and irregular (*n* = 4). The diabetologists who did not declare VD supplementation attributed their decision to a lack of motivation (*n* = 4), absentmindedness (*n* = 1), lack of the need (*n* = 1), “lack of the Polish studies and guidelines” (*n* = 1), doubts about the current supplementation guidelines (*n* = 1). Both groups who declared or denied VD supplementation were compared (Table 2). In the group which declared VD supplementation, there were more physicians who talk about VD supplement use with more than 75% of their patients, and who believe that every medical professional should discuss VD supplementation with the patient, and fewer physicians who do not talk with patients about VD supplementation, do not believe in benefits of VD supplementation, do not recommend VD due to belief of the patients low compliance.

## 4. Discussion

In this cross-sectional survey study, we analyzed the prevalence of VD supplementation among individuals with T1DM, personal reasons, and factors affecting the decision on supplement use. Moreover, we analyzed the association between personal beliefs of diabetologists on VD supplementation and its recommendation to patients.

Previously, it was reported that Danish individuals with both T1DM and type 2 diabetes more frequently supplemented VD than non-diabetic subjects [30]. In the study of Ewers et al., 22% of individuals with T1DM declared VD supplementation [30]. In the more recent study on the Danish adult population, 38% of women and 27% of men supplemented VD in autumn [31]. It was found that 62% of the respondents with T1DM declared VD supplementation, and 57% of them reported the proper dose and period of the supplementation. The largest study on vitamin D status in Poland to date revealed that even two of three Polish citizens suffer from VD deficiency and 19% of the population from VD severe deficiency [4]. That report started a wide discussion in Poland on VD supplementation which could increase awareness among patients and medical professionals. Google Trends data analysis performed by Moon et al. revealed that the interest of Google users in VD increases over time [32]. Moreover, it was observed that in European countries interest of VD peaks during the cold season [32]. Similar trends could be found for Polish Google search engine users in 2004–2019, which could be associated with a high percentage of individuals with T1DM who declared VD supplementation (M.K.—Google Trends data). Another explanation of this high prevalence of individuals supplementing VD is a high proportion of diabetologists who counsel VD to their patients. However, the surveys were collected from a limited population of specialists.

Interestingly, there was no significant difference between men and women with T1DM who declared VD supplementation. VD especially is recommended for individuals at risk of osteopenia and osteoporosis, which is mostly elder women. However, the majority of the study group were young or middle-aged women. Therefore, it may be assumed that the distribution of sex and age in the population does not promote women to be more frequently advised for VD supplementation. 

Approximately, 14% of the responders who declared VD supplementation did not provide the exact dose of supplemented VD. Moreover, a substantial number of entities supplemented VD in very low or very high doses that are not recommended by national guidelines [16]. Therefore, health professionals should educate the patients on proper VD dosage.

In our study, most of the patients who denied VD supplementation admitted a lack of knowledge and a lack of personal motivation to initialize VD supplementation. These justify the continuation of efforts to increase public awareness on VD benefits [33,34]. Moreover, irregular supplementation and a too-low VD daily dose were the most frequent errors in supplementation. To prevent ineffective VD supplementation, medical professionals should motivate their patients for regular and adequate supplementation.

This study provides a unique insight into the determinants of VD supplementation in T1DM population. The physicians’ recommendation was the significant factor of VD supplementation. However, any significant association between pharmacist, relative, or friend recommendation and VD supplementation was not observed. These results may mirror a personal hierarchy of authorities in health issues of the respondents: medical specialists are the most trusted, second are family doctors, while recommendations of pharmacists, relatives, and friends are not considered as reliable. Interestingly, VD supplementation in relatives or friends did not affect the decision on VD supplementation of the respondents. This suggests that encouraging patients for the initialization of VD supplementation might require a more personal approach. Respondents who get to know about VD from Internet/media/books were more eager to supplement VD. Goodman et al. showed that online materials and mobile apps may increase awareness of VD importance [35]. Moreover, Beck et al. reported that 80% of Internet users perceived the Web as a reliable source of health-related information [36]. Since self-education increased the odds of VD supplementation, it is worth considering focusing on the promotion of VD properties online.

Interestingly, living in a village was negatively associated with VD supplementation. However, Zadka et al. reported that parents living in the countryside had higher knowledge of VD and tended to more commonly take VD than respondents living in the city [24]. We hypothesize that individuals with T1DM living in the cities have better access to medical specialists which recommendations are the strong factor affecting the decision on VD supplementation than people living in the countryside.

In most cases, VD supplementation did not have an association with medical history. This may be surprising because it may be suspected that patients with more comorbidities will be more frequently encouraged to take VD for health reasons.

The individuals from Facebook groups were more likely to supplement VD. This may be caused by survey bias. The author spread the links to the digital version of the survey on Facebook groups and encourage the users, while the paper version was mainly handing to the patient. Though, it is possible that active Internet users who are associates of Facebook groups for individuals with diabetes are more aware of the beneficial properties of VD. 

Our survey dedicated to diabetologists revealed that more than three of four declared VD supplementation. The personal decision about VD supplementation was associated with recommending patient VD supplementation. Up to 40% of physicians who supplement VD declared that they recommend VD of most of their patients. Moreover, diabetologists who declared VD supplementation mostly agreed that every medical professional should discuss VD supplementation with the patient. However, 37% of physicians admitted that lack of time limits the possibility to counsel the supplementation. This may mirror the main obstacle for the education of the patient in circumstances of a medical office.

This is, to our best knowledge, the first study investigating the most important personal motivations and factors determining VD supplementation in T1DM population. Furthermore, there were investigated the personal attitude of diabetologists on VD supplementation. We hope that our findings may be an inspiration for a detailed investigation of factors limiting proper VD supplementation in the different local populations.

The authors acknowledge several limitations of the study. Firstly, the data was collected using two different versions of the survey. Secondly, an anonymous survey is associated with the bias of the respondents. The answers came mostly from females with T1DM. Studies suggest that women are more eager to supplement VD than men [30,37]. Taken together, it may be suspected that the number of individuals who declared VD supplementation may be overestimated. Thirdly, the authors focused on T1DM population, which limits the extrapolation of the results to the other populations. Individuals with T1DM should regularly be consulted by a specialist thus may be more exposed to contact with a medical professional who might recommend VD supplementation. Moreover, the progression of the disease leading to diabetic nephropathy and in consequence chronic kidney disease may be an additional indication of VD supplementation. Nevertheless, a modification of the survey could be also used to assess determinants for VD supplementation. The survey for diabetologists did not contain detailed questions on whether physicians assess VD dietary intake and VD status of their patients. In consequence, the study does not provide information on how diabetologists qualify the patients for VD supplementation. Finally, we resigned from the use of the questionnaire assessing VD intake in the diet. These surveys are time-consuming and may require a detailed calculation of consumed food weight which may limit the number of complete responses. Therefore, we cannot conclude whether the VD consumption in the diet was adequate in the study group. Some of the individuals with T1DM may limit the consumption of dairy products due to the high content of fat or high glycemic index. This problem requires further studies.

## 5. Conclusions

Most of the Polish individuals with T1DM use VD supplements and the decision is related to physicians’ recommendations and self-education.

## Figures and Tables

**Figure 1 ijerph-17-00715-f001:**
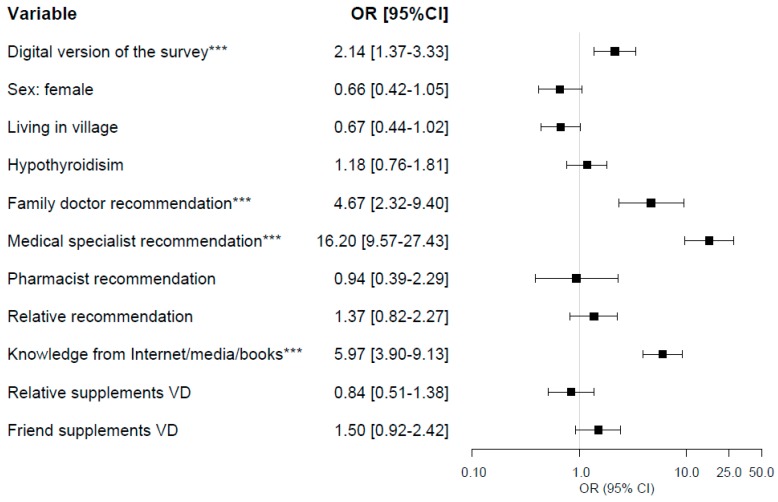
Multivariate logistic regression model. Dependent variable: declaration of vitamin D supplementation; *** *p* < 0.001. CI—confidence interval, OR—odds ratio, VD—Vitamin D.

**Table 1 ijerph-17-00715-t001:** Comparison between respondents who declared or denied vitamin D supplementation. Data presented as median (interquartile range) or number (percentage).

Feature	Total *n* = 734 (100%)	VD Supplementation = YES *n* = 455 (62.0%)	VD Supplementation = NO *n* = 279 (38%)	*p*-Value
**General characteristics and diabetes history**				
**Paper survey**	**184 (25.1)**	**79 (17.4)**	**105 (37.6)**	**<0.001**
Sex: female	551 (75.1)	352 (77.4)	199 (71.3)	0.07
Age [years]	31 (24–39)	32 (24–40)	30 (23–39)	0.05
Age > 65 years	11 (1.5)	5 (1.1)	6 (2.2)	0.26
Diabetes duration [years]	12 (5–20)	12 (5–21)	12 (5–19)	0.55
Weight [kg]	68 (60–79)	68 (59–80)	69 (60–78)	0.63
Height [m]	1.69 (1.64–1.74)	1.68 (1.64–1.74)	1.69 (1.64–1.75)	0.08
BMI [kg/m^2^]	23.9 (21.5–26.7)	24.1 (21.4–26.8)	23.7 (21.6–26.5)	0.68
Overweight (BMI 25–30 [kg/m^2^])	193 (26.3)	116 (25.5)	77 (27.6)	0.53
Obesity (BMI ≥ 30 [kg/m^2^])	83 (11.3)	57 (12.5)	26 (9.3)	0.18
**Living place: village**	**193 (26.3)**	**100 (22.0)**	**93 (33.3)**	**<0.001**
Living place: city < 50,000 citizens	186 (25.3)	110 (24.2)	76 (27.2)	0.35
**Living place: city > 50,000 citizens**	**355 (48.4)**	**245 (53.8)**	**110 (39.4)**	**<0.001**
At least one diabetic complication	156 (21.3)	97 (21.3)	59 (21.1)	0.96
**Retinopathy**	**106 (14.4)**	**54 (11.9)**	**52 (18.6)**	**0.01**
Nephropathy	28 (3.8)	14 (3.1)	14 (5.0)	0.18
Neuropathy	87 (11.9)	57 (12.5)	30 (10.8)	0.47
Diabetic Foot Syndrome	20 (2.7)	10 (2.2)	10 (3.6)	0.26
Ischemic Heart Disease	24 (3.3)	18 (4.0)	6 (2.2)	0.18
**Hypothyroidism**	**226 (30.8)**	**160 (35.2)**	**66 (23.7)**	**<0.01**
Coeliac disease	32 (4.4)	23 (5.1)	9 (3.2)	0.24
Asthma	39 (5.3)	25 (5.5)	14 (5.0)	0.78
**Influence of the respondent’s environment**				
**Family doctor recommendation: YES**	**109 (14.9)**	**96 (21.1)**	**13 (4.7)**	**<0.001**
**Medical specialist recommendation: YES**	**262 (35.7)**	**239 (52.5)**	**23 (8.2)**	**<0.001**
**Pharmacist recommendation: YES**	**53 (7.2)**	**41 (9.0)**	**12 (4.3)**	**0.02**
**Relative recommendation: YES**	**190 (25.9)**	**139 (30.5)**	**51 (18.3)**	**<0.001**
**Friend recommendation: YES**	**148 (20.2)**	**98 (21.5)**	**50 (17.9)**	**0.24**
**Knowledge acquired from Internet/media/books: YES**	**288 (39.2)**	**227 (49.9)**	**61 (21.9)**	**<0.001**
**My relative supplements VD: YES**	**287 (39.1)**	**199 (43.7)**	**88 (31.5)**	**<0.01**
**My friend supplements VD: YES**	**243 (33.1)**	**176 (38.7)**	**67 (24.0)**	**<0.001**
**Number of positive responses [n]**	**2 (1–3)**	**2 (1–4)**	**1 (0–2)**	**<0.001**

VD—vitamin D.

**Table 2 ijerph-17-00715-t002:** Comparison between diabetologists who declared or denied vitamin D supplementation. Data presented as number (percentage) or median (interquartile range).

Variables	Total *n* = 38 (100%)	Declared VD Supplementation *n* = 30 (79%)	Denied VD Supplementation *n* = 8 (21%)	*p*-Value
Sex: female [*n*]	28 (74)	20 (67)	8 (100)	0.057
Practice [years]	26 (18–31)	29 (17–32)	20 (18–23)	0.21
Percentage of patients which talk about VD supplementation [*n*]:				
**>75%**	**12 (32)**	**12 (40)**	**0 (0)**	**0.03**
25–75%	13 (34)	10 (33)	3 (38)	0.83
<25%	9 (24)	7 (23)	2 (25)	0.92
**0%**	**4 (11)**	**1 (3)**	**3 (38)**	**<0.01**
Agree with the sentence				
“I have not enough time to counsel VD supplementation”	14 (37)	10 (33)	4 (50)	0.39
**“I believe that VD supplementation does not provide significant benefits to my patients”**	**4 (11)**	**0 (0)**	**4 (50)**	**<0.001**
**“I think that is makes no sense to recommend VD supplementation because patients will not use it regularly”**	**1 (3)**	**0 (0)**	**1 (13)**	**0.049**
“I do not recommend VD supplementation to not overload the patient with additional costs”	0 (0)	0 (0)	0 (0)	-
**“I believe that every medical professional should discuss VD supplementation with the patient”**	**32 (84)**	**28 (93)**	**4 (50)**	**<0.01**

VD—vitamin D.

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
