# Peer review of "Determinants of Vitamin D Supplementation among Individuals with Type 1 Diabetes"

_ijerph, 2020, doi:10.3390/ijerph17030715_

Round 1

Reviewer 1 Report

This study aimed to identify factors associated with vitamin D supplement use in individuals with type 1 diabetes. However, some precisions to the paper could be helpful.

General comments:

How was the questionnaire developed? Was it based on any framework, previous findings, studies in other populations, specialists’ opinions? More information regarding the development of this questionnaire and the choice of determinants included should be included in the manuscript. Was the questionnaire pre-validated or tested in a subset of individuals?

Overall, for the questionnaire, more details should be included in the methods. It is unclear how some of the determinants were defined and assessed.

Also for the questionnaire, if it was the same questionnaire for the paper version and the web-based version, why include the type of questionnaire in the analyses? Using a web-based version only could have possibly led to some bias in the respondents by limiting the possibility to some individuals to participate in this study, but it seems irrelevant in this case to include the type of survey in the multivariate analysis (the type of survey can’t possibly be a determinant of VD supplement use). It seems like authors could mention they offered both a paper and a web-based version of the questionnaire, but the type of questionnaire should not be included in the analysis.

Throughout the manuscript, authors sometimes use VD intake where they seem to mean “VD supplement use”. Intake would also refer to intake via food while authors are referring to supplementation.

For the diabetologists survey, authors stratified by whether the specialists used VD supplements themselves? Can authors explain why? Why not simply include it as one of the variables?

Abstract

Line 10: Consider changing “suffer from vitamin D deficiency” for “present vitamin D deficiency”.

Line 18: In the two sentences, authors mention “VD intake”, do they refer to “VD supplement intake”?

Lines 23-24: This conclusion simply rephrases the previous sentence. Perhaps authors could provide a more general conclusion in this context?

Introduction

Page 2 Line 45: would recommend changing “risk factors” for “potential risk factors”

Page 2 Line 53: The sentence beginning by “However” is not quite clear, please rephrase.

Methods

Page 2 Lines 62-63: Is it recommendations from friends or use of supplements by friends? This sentence is unclear.

Page 2 Line 63: How was knowledge on VD assessed and defined?

Page 2 Line 64: How was personal motivation assessed and defined?

Page 2 Line 65: What does “4.5-6.5 postgraduate specialization training” refer to?

Page 2 Line 67: Why include participants starting at 16? Wouldn’t their supplement use also be influenced by parental beliefs, parents’ will to buy the supplements etc.? Are the findings similar if only adult participants are included?

Results:

Page 4 Lines 127-134: As mentioned above, it is not quite surprising that respondents to the paper and the web questionnaires are different, but since it is the same questionnaire, it seems unclear why it would be necessary to compare these groups based on the format of the questionnaire.

Page 4 Line 142: “resignation” or “non-initiation”? Did authors include questions regarding past VD supplement use or only current VD supplement use?

Discussion:

Page 8 Line 265: “Firstly” is used in two subsequent sentences.

Conclusion:

The conclusion is simply a mention of the main findings, authors could perhaps expand their conclusion with interpretation and perspectives.

Table S1: Is part of the questionnaire missing? This seems to include only general characteristics questions

Reviewer 2 Report

In Europe, independently from the country, the deficiency of vitamin D is reported in the case of even 30% of adults. Vitamin D not only influences on bone metabolism, but also modifies the immune response and tissue sensitivity to insulin, which can be important for people with type 1 diabetes. Therefore, the assessment of vitamin D supplementation in this group of patients seems particularly important. However this assessment was not accompanied by an evaluation of nutritional status of this vitamin and determination of its consumption with the diet. Knowledge of the concentration of vitamin D in the blood of patients participating in the study could indicate the desirability of using its supplementation. These values could also be referred to the doses of supplementation that were used.

In spite of the vitamin D synthesis mediated by UVB radiation, the sunshine avoiding behaviors cause that in European countries the dietary intake is becoming even more important source of vitamin D. Assessment of vitamin D intake with diet is an important complement to information about its supplementation and together with it gives a full picture of its intake. The assessment of vitamin D intake is necessary to further assess the advisability of supplementation. There are several frequency questionnaires of vitamin D intake that have been validated for the United States of America, Canada, as well as for European countries such as Finland, Sweden, Great Britain, Ireland and Poland.

The authors did not include in the supplement this part of the questionnaire which was devoted to vitamin D supplementation by patients with type 1 diabetes. The questionnaire addressed to diabetologists did not include questions about recommendations for assessing the nutritional status of vitamin D in patients with type 1 diabetes (e.g. determination of blood vitamin D concentration) and assessing dietary vitamin D intake. In the survey for doctors, there is also no question about the reason for vitamin D supplementation, and just like in the survey for patients, questions about the intake of food sources of this vitamin. Answers to these questions would justify the use of vitamin D supplements and doses of this supplementation. Only 4% of patients declared that they were supplementing vitamin D due to abnormal laboratory results.

In the multivariate regression analysis, the authors did not notice the significant impact of the recommendations of pharmacists or relatives on vitamin D supplementation, but it is worth noting that these factors were significant, comparing their independent effects in people with type 1 diabetes who declared vitamin D supplementation with patients who did not apply such supplementation (in univariate logistic regression analysis).

Description of vitamin D doses used by patients is not well understood - the given values do not add up to 100%.There is also no explanation for the reasons why a high percentage of type 1 diabetes patients use very high doses of vitamin D supplementation (13,000 to 80,000 UI/d).

The authors rightly say that doctors should motivate their patients to regularly and appropriately adjusted vitamin D supplementation, but they should also pay attention to recommendations regarding the proper intake of this vitamin with diet. Some patients with type 1 diabetes eliminate, for example, dairy products from the diet.

The characteristics of the group of diabetologists participating in the survey can be transferred to the research methodology.

In the first paragraph of the results there is information about the age range: 24-39 (line 108), however, as it follows from the further part of the chapter, this is the interquartile range - wouldn't it be better to give a range of values from the lowest to the highest values?

Round 2

Reviewer 1 Report

The mansucript as revised by authors has improved and the methods are clearer. Here are a few minor remaining suggestions:

Abstract

Line 16: change “the most important personal motivation on VD supplementation” for “the most important personal motivator for VD supplement use”

Introduction

Lines 53-54: Change “However, there is little known on factors deciding on VD supplementation.” To “However, there is little known on factors leading to the decision to use VD supplements or not”.

Results

Line 152: In the subtitle “3.3. Personal reasons for or against Vitamin D supplementation”, the wording suggests participants were in favor or not of supplementation, while the questions reflect their use or not. I suggest changing the subtitle to “3.3. Personal reasons to use Vitamin D supplementation or not
